# Recurrent Differentiated Thyroid Cancer: The Current Treatment Options

**DOI:** 10.3390/cancers15102692

**Published:** 2023-05-10

**Authors:** Andrés Coca-Pelaz, Juan Pablo Rodrigo, Jatin P. Shah, Iain J. Nixon, Dana M. Hartl, K. Thomas Robbins, Luiz P. Kowalski, Antti A. Mäkitie, Marc Hamoir, Fernando López, Nabil F. Saba, Sandra Nuyts, Alessandra Rinaldo, Alfio Ferlito

**Affiliations:** 1Department of Otolaryngology, Hospital Universitario Central de Asturias, University of Oviedo, ISPA, IUOPA, CIBERONC, 33011 Oviedo, Spain; jprodrigo@uniovi.es (J.P.R.); lopezafernando@uniovi.es (F.L.); 2Head and Neck Service, Department of Surgery, Memorial Sloan Kettering Cancer Center, New York, NY 10065, USA; shahj@mskcc.org; 3Department of Surgery and Otolaryngology, Head and Neck Surgery, Edinburgh University, Edinburgh EH3 9YL, UK; iain.nixon@nhslothian.scot.nhs.uk; 4Department of Otolaryngology-Head and Neck Surgery, Institut Gustave Roussy, CEDEX, 94805 Villejuif, France; dana.hartl@gustaveroussy.fr; 5Laboratoire de Phonétique et de Phonologie, 75005 Paris, France; 6Department of Otolaryngology-Head and Neck Surgery, Southern Illinois University School of Medicine, Springfield, IL 32952, USA; kthomasrobbins@gmail.com; 7Head and Neck Surgery and Otorhinolaryngology Department, A C Camargo Cancer Center, São Paulo 01509-001, Brazil; lp_kowalski@uol.com.br; 8Department of Otorhinolaryngology, Head and Neck Surgery, Research Program in Systems Oncology, Faculty of Medicine, University of Helsinki and Helsinki University Hospital, FI-00014 Helsinki, Finland; antti.makitie@helsinki.fi; 9Department of Head and Neck Surgery, UC Louvain, St Luc University Hospital and King Albert II Cancer Institute, 1200 Brussels, Belgium; marc.hamoir@saintluc.uclouvain.be; 10Department of Hematology and Medical Oncology, The Winship Cancer Institute, Emory University, Atlanta, GA 30322, USA; nfsaba@emory.edu; 11Laboratory of Experimental Radiotherapy, Department of Oncology, Leuven Cancer Institute, University Hospitals Leuven, 3000 Leuven, Belgium; sandra.nuyts@uzleuven.be; 12Department of Radiation Oncology, Leuven Cancer Institute, University Hospitals Leuven, 3000 Leuven, Belgium; 13ENT Unit, Policlinico Città di Udine, 33100 Udine, Italy; dottalerinaldo@gmail.com; 14Coordinator of the International Head and Neck Scientific Group, 35100 Padua, Italy; profalfioferlito@gmail.com

**Keywords:** differentiated thyroid cancer, salvage surgery, radioiodine therapy, targeted therapy, tyrosine kinase inhibitors

## Abstract

**Simple Summary:**

The most appropriate treatment for local/regional recurrent differentiated thyroid cancers (DTC) is surgical excision if feasible, followed by adjuvant radioiodine administration. However, depending on the location or extent of the recurrence, surgical excision may not be possible, thus creating a need for alternative therapeutic interventions to maintain the otherwise excellent prognosis for this disease. In light of the increased incidence of this disease and the plethora of available systemic options, a detailed knowledge of the available treatment options is essential for successful management of recurrent DTC.

**Abstract:**

Differentiated thyroid carcinomas (DTC) have an excellent prognosis, but this is sometimes overshadowed by tumor recurrences following initial treatment (approximately 15% of cases during follow-up), due to unrecognized disease extent at initial diagnosis or a more aggressive tumor biology, which are the usual risk factors. The possible sites of recurrence are local, regional, or distant. Local and regional recurrences can usually be successfully managed with surgery and radioiodine therapy, as are some isolated distant recurrences, such as bone metastases. If these treatments are not possible, other therapeutic options such as external beam radiation therapy or systemic treatments should be considered. Major advances in systemic treatments have led to improved progression-free survival in patients previously considered for palliative treatments; among these treatments, the most promising results have been achieved with tyrosine kinase inhibitors (TKI). This review attempts to give a comprehensive overview of the current treatment options suited for recurrences and the new treatments that are available in cases where salvage surgery is not possible or in cases resistant to radioiodine.

## 1. Introduction

This article was written by members and invitees of the International Head and Neck Scientific Group (http://www.IHNSG.com).

Differentiated thyroid carcinoma (DTC), which includes papillary and follicular cancers, comprises greater than 90% of all thyroid malignancies [1]. Improved knowledge of the biology of these lesions has led to an improvement in their management following recognized clinical guidelines, leading to an excellent prognosis for the majority of patients with DTC.

Depending on the definition used for recurrent disease (biochemical or structural) and the distinction between persistent and recurrent disease, the incidence of recurrent DTC varies significantly. Differentiating between a persistent and recurrent tumor is somewhat arbitrary and is usually determined by the initial surgical resection with no residual tumor (R0), the presence of microscopic residual tumor (R1), or gross residual disease (R2) [2]. However, from a practical view, it is likely that all DTC recurrences are the result of either microscopic or macroscopic persistent disease [2,3].

Most patients (>90%) can be classified as stage I or II using the TNM classification in order to estimate their risk of cancer-related death [4,5]. The American Thyroid Association (ATA) risk stratification criteria can estimate the hazard of structural recurrence. Based on this risk classification, DTC recurrence is not uncommon, occurring in 3–13% of low-risk patients, 21–36% of intermediate-risk patients, and approximately 68% of high-risk patients [3]. Applying these three categories, most patients (80%) are classified as low risk.

Most of the parameters used in the TNM staging and ATA prognostic classifications can be determined prior to the initial treatment and thus are used to guide appropriate treatment [3]. In addition, risk recalculation must be done post-treatment using the “ongoing risk reassessment” system that classifies patients into four groups: excellent response, biochemically incomplete response, indeterminate response, and structurally incomplete response [3]. Persistent disease can be differentiated from recurrent disease using this dynamic risk stratification system. Only patients previously classified as excellent response can be cataloged as disease recurrence [6].

All of these parameters and classifications play an important role because despite the worldwide increasing incidence of DTC, over the past two decades, the mortality rate has remained stable or only slightly decreased, supporting a less aggressive therapeutic approach [6,7]. For this purpose, investigation of more precise predictors of the risk of recurrence and survival is essential.

During follow-up, the tools applicable for detecting a possible recurrence are serum thyroglobulin (Tg), serum anti-Tg antibodies, high-resolution ultrasonography (US) for thyroid bed and neck recurrences, as well as computed tomography (CT), magnetic resonance imaging (MRI), and positron-emission tomography (PET-CT) for neck and distant metastases. Radioactive iodine (RAI) scan can potentially facilitate the detection of low-volume disease in differentiated carcinomas. In patients with positive findings of structural disease, it is mandatory to perform cytological fine-needles aspiration (FNA) biopsy, which can be complemented by Tg measurement in the needle wash-out for confirmation of tissue diagnosis [8,9]. The location, extent, and trajectory of recurrent disease will determine the most appropriate salvage treatment: active surveillance, surgery, RAI treatment, external beam radiotherapy (EBRT), image-guided techniques (ethanol injection and radiofrequency ablation), or systemic therapy. There is, however, a lack of standardized approaches for surveillance.

Given the advances in treatment for DTC, the aim of this article is to review the treatment options available for each type of recurrence, in order to highlight and prioritize all of the options for the decision-making process.

## 2. Types of Recurrences

Recurrences in DTC can be divided in two types: neck/upper mediastinum (loco regional) or distant. Within the cervical region, we have to differentiate between thyroid bed recurrences or residual thyroid tissue, and lymph node recurrences (central or lateral) [10]. Recurrent DTC can also be divided, as stated above, into biochemical recurrence (abnormal Tg or rising anti-Tg antibody levels in the absence of localizable disease), which carries an excellent prognosis; imaging-detected recurrence, which is a structurally recurrent disease (persistent or newly identified loco-regional or distant metastases); or clinical recurrence (detected on physical examination). Obviously, clinical recurrences are most threatening to both overall disease control and long-term survival [2]. Here, we review each type of recurrence and its treatment.

Biochemical recurrences are defined as recurrences occurring in patients with negative imaging and suppressed Tg ≥ 1 ng/mL or stimulated Tg ≥ 10 ng/mL (both in the absence of anti-Tg antibodies) or rising anti-Tg antibody levels. This kind of recurrence spontaneously evolves into no evidence of structural disease (NED) or is controlled after additional therapy in 55.6% of the patients, while 44.4% remain with persistent disease or go on to develop structural disease. Less than 1% of these patients die from their disease [11]. In patients with a biochemical recurrence (rising levels of Tg or anti-Tg), further imaging studies should be performed to locate a possible structural recurrence justifying additional treatment. In the absence of structural disease, continued monitoring is warranted and is considered sufficient [3].

A structurally incomplete response is defined as clinical, radiological (US, CT), or functional evidence of disease (RAI scan, 18-FDG-PET) occurring as loco-regional or distant metastases, with any Tg level and with or without anti-Tg antibodies. Most patients (50–85%) with a structurally incomplete response will have physical, imaging, and/or biochemical evidence of persistent disease at last follow-up despite additional therapy. In contrast with biochemical recurrences, death from disease has been reported in 11% of patients with a loco-regional incomplete response and in 57% of patients with anatomically identifiable distant metastases [4,11,12]. However, these results were reported prior to the approval of TKIs in the management of DTC. It must be noted, however, that in a select group of patients with distant metastases, a long survival has been reported [13]. As expected, structural persistence or recurrences at the neck are more likely to respond to further treatment and therefore will have a lower disease-specific mortality rate compared to patients with distant recurrences. A structural recurrence in this setting therefore requires an active therapeutic approach. The structural disease must be clearly defined on anatomical studies, which usually include US, axial contrast-enhanced CT, and/or contrast-enhanced MRI.

An intermediate category designated “indeterminate response” indicates borderline biochemical, equivocal structural, or functional findings that exclude both an excellent response and definitive persistent disease. This group of patients have a prognosis that is intermediate [14,15,16]. It includes patients with small avascular thyroid bed nodules, non-biopsied atypical cervical lymph nodes, an area of weak uptake in the thyroid bed associated with non-stimulated Tg values < 1 ng/mL, TSH-stimulated Tg values between 1 and 10 ng/mL, and so on. Between 12 and 29% of ATA low-risk patients, 8–23% of ATA intermediate-risk patients, and 0–4% of ATA high-risk patients are classified as indeterminate response [11,15]. In the long-term follow-up, the majority of these patients remain disease-free, although slightly more than 20% may have biochemical, functional, or structural evidence of disease progression and would require additional therapies [3].

## 3. Initial Treatment

Independent of ATA risk status, surgery represents the first line of treatment for most patients with DTC, although the extent of surgery is dependent on preoperative factors. In low-risk patients, surgery includes total thyroidectomy or unilateral lobectomy. While therapeutic selective neck dissection (levels IIa, III, IV, Vb, and VI) is indicated in patients with documented lymph node metastases, elective dissection of the central and lateral compartments is not indicated in low-risk patients [3,8,17]. The administration of ^131^I after surgery in low-risk patients is generally not indicated and should be considered on a case-by-case basis with the aim of administering the lowest effective dose since its influence on prognosis in this group has not been proven. Other post-surgical factors such as elevated serum Tg levels may aid in decision making [18,19]. For patients with intermediate risk, depending on the definition used, a lobectomy is recommended for unilateral intrathyroidal disease, but for high-risk patients, surgery should include total thyroidectomy and elective central compartment neck dissection. The surgery in this setting should be an extracapsular total thyroidectomy, leaving no residual thyroid tissue behind, thus avoiding the need “remnant ablation”. The indication in this setting for ^131^I treatment would be persistently elevated Tg to eradicate persistent neoplastic foci. The dose of ^131^I delivered is dependent on the degree of risk [20].

## 4. Neck Recurrences

Following first-line treatment for DTC, the lymph nodes of the neck are the most frequent site of recurrence [8]. However, the presence of gross extrathyroidal extension at the time of surgery is also an important factor for recurrence in the thyroid bed. The histological findings of R0 (no residual tumor), R1 (microscopic residual tumor), or gross residual disease (R2) resections are used to classify patients into persistent or recurrent tumor [2]. After initial treatment, if the patient has gross disease, it is known that local recurrence is more frequent, especially for those who present with T4a tumors involving trachea, esophagus, or recurrent laryngeal nerve or bulky metastatic nodes [2]. It is also important to consider the risk of recurrence for selected patients with more aggressive histologies (insular, solid, tall cell, etc.) [21], advanced age, and BRAF or other mutations [22,23]. Patients with a neck recurrence must undergo a comprehensive evaluation relying on imaging studies for loco regional disease as well as distant metastases.

When suspicious but small sub-centimeter lymph nodes are detected on US, they can be observed for progression. A recent retrospective study [24] found a 51-month progression-free survival (PFS) rate in 14 patients out of 40 (35%) who were followed with observation for metastatic lymph nodes < 2 cm at their largest diameter. The ATA guidelines suggest that central compartment lesions larger than 5 mm (in the smallest diameter) and lateral neck lesions larger than 8 mm (in the smallest diameter) may be submitted for FNA biopsy (with Tg determination on aspiration if needed), but only if a specific treatment is planned [3].

Lymph node metastases can be treated with surgery with or without post-operative ^131^I therapy or only surgery in the case of ^131^I refractory disease. If the recurrence is located at a neck level that has not been previously dissected, compartmental dissection of the targeted level is indicated. However, if the targeted neck level has already been dissected, selective lymph node excision can be performed [25]. In such cases, the use of intraoperative probe may facilitate the localization of the suspicious lesion and its complete resection [26]. In a study of patients with clinical neck recurrence from papillary thyroid cancer, Wang et al. [27] found that isolated nodal recurrence developed in only 3% of patients. These patients were treated with salvage surgery and adjuvant therapy, resulting in a 5-year disease-specific survival of 97.4%. Alternately, for limited lesions (less than 2 cm) that show RAI uptake, ^131^I can serve as an alternative treatment for recurrence [28,29]. Other treatments that have been used for cases of isolated metastasis include ethanol injection and US-guided percutaneous ablation (radiofrequency, laser, microwave, or cryotherapy) [30,31,32]. Retrospective studies show variable rates of successful treatment of neck recurrences, with an excellent response attained in 17–100% of patients and most studies reporting a rate of 65–75% [29,33,34,35,36,37,38,39,40,41,42,43]. After surgery for a recurrence, the use of ^131^I should be considered adjunctively if RAI-avid residual disease is known or suspected. The discordant fact about this attitude is that there are no studies that have shown an improvement of prognosis after RAI therapy. In the case that the diagnostic scan is negative, it has been seen that between 20 and 64% of patients have a positive post-treatment scan. In addition, more than half of the patients with a negative diagnostic scan who were also treated showed a drop in serum Tg levels, although no improvement in survival could be observed [3,44].

The presence of extensive unresectable neck recurrence can be fatal. Such recurrences may involve the larynx, trachea, esophagus, or adjacent major vessels [45]. As suggested by Hartl et al. [46], surgery, if feasible, should be performed to avoid local complications (bleeding, fistulas, etc.) for patients who may otherwise be candidates for treatment with tyrosine kinase inhibitors (TKI). TKI has also been used as a neoadjuvant agent prior to salvage surgery [17,47,48,49].

After surgery for recurrence (to eradicate thyroid remnants as well as incompletely resected DTC) or when salvage surgery is not feasible (non-resectable DTC), in low-volume disease, RAI is a treatment option that should be considered. Postoperative RAI remnant ablation for DTC is also intended to ablate persistent microscopic disease. In a systematic review and meta-analysis including 10 studies with 3821 patients, James et al. found no differences in comparing the long-term recurrence rates (RR, 0.88; 95% CI, 0.62–1.27, *p* = 0.50) and successful remnant ablation (RR, 0.95; 95% CI, 0.87–1.03; *p* = 0.20) for low-dose versus high-dose RAI. They concluded that low-dose (activity) RAI is preferable to high-dose in low- and intermediate-risk DTC. With similar efficacy, low-dose treatment is preferred owing to its lesser side effects [50].

In cases of unresectable or non-iodine-avid neck disease, or localized distant metastatic disease, another option is EBRT [3,51]. In 2016, So et al. [51] published a retrospective review of prospective data from an Australian institution, with 32 treated patients with EBRT. The mean radiation dose was 57.0 Gy (range: 50–63 Gy), with the mean number of fractions being 28 (range: 25–30 fractions). Four out of six patients treated with radiation were free of disease or progression, while 13 of 16 patients who received locoregional EBRT were asymptomatic during the follow-up. The most frequent distant metastatic site was the bone, with a total of 45 sites irradiated. Of these patients, 93% were symptom-free at two years and 78% at four years. They concluded that EBRT can provide durable locoregional control and appears to provide a good palliative option for locoregional and bony metastatic disease.

In cases of symptomatic metastases or when there is a high risk of local complications, stereotactic radiation to treat isolated metastatic lesions, or thermal ablation if feasible, should be considered prior to initiation of systemic therapy [3].

The last option available for neck recurrences is systemic therapy with a TKI, such as sorafenib, pazopanib, sunitinib, lenvatinib, axitinib, cabozantinib, and vandetanib [3,52]. They should be considered in cases of rapidly progressive disease or symptomatic cases, concurrent metastatic disease, and/or when life threatening disease that is not otherwise amenable to local control (Figure 1a), and where surgery would result in significant morbidity such as cases of laryngotracheal or esophageal resection (Figure 1b), where TKIs might downstage in a kind of neo-adjuvant approach to recurrence [53,54]. Evidence of the effects of these treatments on overall survival (OS) are lacking given the cross-over allowed on randomized clinical trials. The significant improvement in PFS is an indicator of the excellent clinical activity of these agents; lenvatinib when compared to a placebo in ^131^I-refractory thyroid cancer was associated with improvement in PFS and disease response rate [55]. Similarly, sorafenib significantly improved PFS compared with placebo in this kind of patient [56]. Further investigations of quality of life and economic implications with the use of these agents are needed [57].

However, side effects of these treatments are reported in up to 20% of patients, leading to dropping out of treatment and frequent dose reductions, necessitating careful selection and monitoring [3]. Therapy-related deaths reported in a meta-analysis by Schutz et al. had an incidence of 1.5% (95% CI, 0.8% to 2.4%) with a relative risk of 2.23 (95% CI, 1.12 to 4.44; *p* = 0.023) when compared to control patients [58]. Although the risk of therapy-related deaths is low, given the high rate of adverse effects of TKI treatments, a crucial factor in the decision to administer TKIs will be whether to administer these treatments in a patient who is asymptomatic, with stable or slowly progressive disease. In this regard, the ATA published a set of factors for and against administering TKI treatments. Suitable candidates should have progression of disease, presence of disease symptoms, and evidence of disease dissemination. Contraindications include liver malfunction, intestinal disease, recent bleeding, and cardiovascular events, or contraindications to VEGF TKIs [3].

## 5. Distant Recurrence

Although DTC generally has a good prognosis, a small subset of patients will develop distant metastasis. Overall, distant disease has been reported to occur in 4–21% of patients [59,60,61,62]. The predominant sites of metastases are pulmonary and osseous. Less common sites include the liver, brain, skin, ovaries, and adrenal glands [8,59,60]. Bone metastases can bring excessive morbidity and poor quality of life for these patients, while pulmonary metastases are the commonest cause of death due to respiratory failure [63]. Aziz et al. [60] published a retrospective, single-center study with 117 patients who had metastatic DTC. They concluded that the most significant factors in predicting the outcome in metastatic DTC are age, extra-thyroidal extension, and distant metastasis. Zhong et al. [59] studied the impact of lung metastasis compared with bone, brain, or liver metastasis. Included were various types of thyroid carcinoma identified in a Surveillance, Epidemiology, and End Results (SEER) database of 77,322 patients, in which the lung was the most prevalent site. Like brain metastasis, the outcomes of patients with lung metastasis are worse than the outcomes for those with liver or bone metastasis. Multi-organ metastases patients had the worse survival outcome.

In the case of low-volume or miliary lung metastasis, patients should be treated with RAI therapy as long as the lesions concentrate RAI and a clinical response is seen, repeating treatment every 6 to 12 months [3]. In the case of pulmonary macrometastasis, clinical remission with RAI therapy is uncommon, and the prognosis is poor, but in selected cases, surgical resection should be considered if feasible [64]. In a study by Cho et al. [65], the only independent factor for a poor prognosis in patients with lung metastasis was non-avidity for ^131^I, which occurs more frequently in late metastasis (diagnosed by whole-body scan or radiologic examination performed due to increases in T4-suppressed Tg or stimulated (off T4) Tg after initial evaluation or ^131^I remnant ablation).

Treatment for bone metastasis with RAI therapy can provide symptomatic benefits to patients but is rarely curative. In cases of isolated metastasis or in oligometastatic disease that are going to receive RAI therapy, they must be evaluated for local therapy of lesions when metastases are progressive, symptomatic, or at high risk of causing symptoms or complications (Figure 2). Treatment options include surgery (stabilization with or without partial tumor resection), metastasectomy (complete resection of the tumor), radiotherapy (EBRT, stereotactic radiosurgery, or intensity-modulated radiotherapy), and percutaneous procedures such as thermal ablation techniques (radiofrequency ablation, laser ablation, cryotherapy), cementplasty, or embolization [3,9,66,67].

Finally, there are systemic therapies such as TKIs and antiresorptive therapies [68,69]. Bonichon et al. [70] investigated the different treatments for DTC metastases and their indications. They found that despite new targeted therapies, local treatment still has an important role, which is either palliative or potentially curative for oligometastatic lesions. Even in the case of extensive disease, it may allow delaying of the initiation of TKI therapy, which is expensive, must be continued life-long, and is associated with significant side effects. However, TKIs offer a potentially improved outcome for patients with distant metastasis and are indicated in patients with progressive RAI-refractory disease and with a high tumor burden [3,32].

Some studies, such as the one by Cabanillas et al. [71], found that the most noticeable responses to sorafenib and sunitinib occurred in the lungs, in contrast to minimal changes in nodal metastasis and progression of the disease in pleural and non-irradiated bone metastases. This suggests that multimodal approaches with combined local and systemic therapies may be more plausible in this clinical scenario.

Antiresorptive therapies, such as bisphosphonates, are used for preventing skeletal-related events in patients with bone metastasis. Bisphosphonates inhibit osteoclast-mediated bone resorption and inhibit tumor cell activity by impairing angiogenesis and inducing apoptosis [72]. Denosumab, a monoclonal antibody targeting the receptor activator of nuclear factor-kappa B ligand (RANKL), which inhibits osteoclast function, has been used in patients with bone metastasis. It has proven to be superior to bisphosphonates in the prevention of skeletal-related events but does not impact OS [73].

In the case of brain metastasis, the preferred treatment options are surgical resection and stereotactic radiotherapy [3]. The usefulness of stereotactic radiosurgery was analyzed in an international multicenter study by Bunevicius et al. [74], who found that this treatment allows for durable local control in the vast majority of patients. The worst OS was for patients with multiple brain metastasis and compromised functional status (lower Karnofsky performance score) at the time of the treatment. If patients need treatment with RAI, in order to reduce the effects of a TSH-induced increase in tumor size, as well as the inflammatory response induced by the RAI or EBRT, stereotactic radiotherapy and concomitant glucocorticoid therapy should be used [3,75].

## 6. New Perspectives

In recent years, research has continued to investigate the different treatment options for DTC patients (especially for those refractory to RAI) or for improving the tolerance of systemic therapies. Brose et al. [76] published data regarding the phase 3 COSMIC-311 trial of 258 patients (170 treated with cabozantinib and 88 with placebo) who received cabozantinib for previously treated RAI-refractory DTC. The median PFS was 11.0 months (96% CI, 7.4–13.8 months) for cabozantinib and 1.9 months (96% CI, 1.9–3.7 months) for the placebo group (hazard ratio, 0.22; 96% CI, 0.15–0.32; *p* < 0.0001). One patient had a complete response from cabozantinib, and the objective response rate was 11.0% (95% CI, 6.9–16.9%) versus 0% (95% CI, 0.0–4.1%; *p* = 0.0003). The authors confirmed that cabozantinib was associated with a higher disease stabilization rate and a higher rate of reductions in target lesions. OS favored cabozantinib, but interpretation is limited by the crossover of patients between the groups. Busaidy et al. [77] published the results of an open-label randomized phase 2 multicenter trial with BRAF-mutated RAI-refractory DTC with progressive disease. They compared the use of dabrafenib vs. dabrafenib and trametinib and concluded that the combination was not superior in efficacy to dabrafenib alone for such patients. Dabrafenib’s most common treatment-related adverse events included skin and subcutaneous tissue disorders (17/26, 65%), fever (13/26, 50%), and hyperglycemia (12/26, 46%), whereas nausea, chills, and fatigue occurred in 16/27 (59%) for dabrafenib alone versus 14/27 (52%) for the combination of dabrafenib and trametinib.

Toxicity is often a limiting factor with systemic treatments for DTC. In this regard. Taylor et al. [78] performed a multicenter, randomized, double-blind phase 2 study, comparing patients with an initial dose of lenvatinib of 18 mg/day to patients with a starting dose of 24 mg/day (approved starting dose). Health-related quality of life was studied with EQ-5D and FACT-G scores. They concluded that health-related quality of life was not statistically different from that of patients in the 24 mg/day arm, thus supporting the use of the lenvatinib with the starting dose of 24 mg/day, since this dose does not increment toxicity measured in terms of quality of life.

Kiyota et al. [79] analyzed the data from a previously published phase 3 international, randomized, double-blind, multicenter trial (SELECT), in order to know the impact of tumor burden at baseline and tumor response on OS in patients with RAI-refractory DTC treated with lenvatinib. They suggested that a possible prognostic marker of OS is the sum of diameters of the target lesions in patients receiving lenvatinib, since a lower tumor burden is related with a prolonged OS compared with patients with a higher tumor burden. These observations support the concept of early initiation of lenvatinib treatment.

Brose et al. [80] published an international, prospective, open-label, non-interventional cohort study in order to know when to start treatment with TKIs (sorafenib and lenvatinib) in patients with RAI-refractory DTC. There were 647 patients with a median duration of observation of 35.5 months (range < 1–59.4). There were 344 TKI-treated patients (209 received sorafenib, 191 received lenvatinib, and 19 received another TKI at some point). There were two cohorts: patients for whom a decision to initiate TKI was made at study entry and patients for whom it was decided not to start such treatment. Time to symptomatic progression was 55.4 months overall, 55.4 months in patients whose treatment was initiated at study entry, and 51.4 months in the other group. Time to symptomatic progression was ≥36 months in 64.5% of patients overall, 59.5% of patients in the first group, and 66.4% of patients in the second group. Median PFS from the start of TKI therapy was 19.2 months, and from the start of sorafenib therapy, 16.7 months. The results of this study suggest that the timing of initiation of TKI treatment does not significantly influence patient outcome.

There are studies that are trying to find a role of immunotherapy agents such as pembrolizumab (immune checkpoint inhibitor targeting PD-1 on immune cells) as a single agent or combined with lenvatinib [81], as well as with the use of RET inhibitors such as selpercatinib [82].

Another aspect of the guidelines that should be addressed in the future is the dilemma that exists regarding whether patients ought to be started on a broad-spectrum TKI versus a genomically targeted agent (in case they have BRAF mutation or NTRK fusion, for example).

## 7. Conclusions

Despite the generally good prognosis of DTCs, there are cases of recurrences that can compromise patient survival. The approach to treatment for such recurrences (Figure 3) should be based on available guidelines. Most cervical recurrences can be successfully managed with surgery, RAI therapy, and in selected cases with image-guided ablation techniques. In cases where these treatments are not possible, other tools such as EBRT or systemic therapies should be used. In the case of distant recurrences, management may also require the use of different treatment modalities drawing on expertise from beyond the traditional thyroid multidisciplinary team, combining approaches to obtain the best results.

## 8. Future Directions

In trying to reduce the risk of recurrences as much as possible, it is essential that the first treatment is appropriate and follows the available guidelines. When recurrence develops, it is important that the physicians involved in this treatment are aware of the available treatments for each possible location of recurrence, in order to ensure that the treatment is appropriate and does not compromise patient survival. Complex recurrences should be sent to units with a high volume of cases. In future, the treatment of recurrent DTC will likely be based on an understanding of the molecular biology of these tumors, and the development of systemic therapy that can effectively treat recurrences when standard treatment modalities are no longer effective.

## Figures and Tables

**Figure 1 cancers-15-02692-f001:**
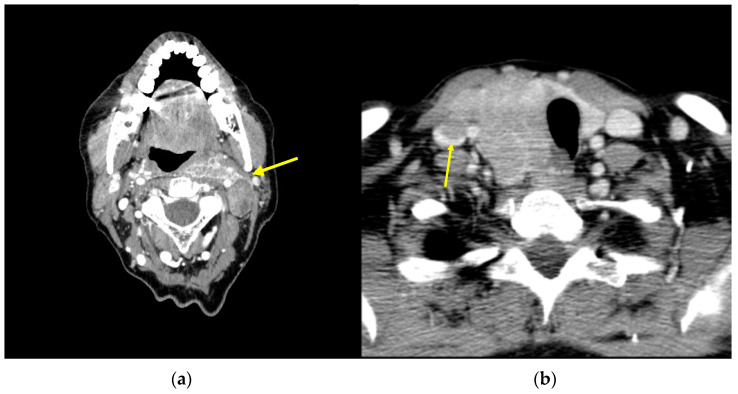
Axial CT scan image: (**a**) neck recurrence (arrow) of DTC affecting the great vessels; (**b**) DTC infiltrating the esophagus and internal jugular vein (arrow).

**Figure 2 cancers-15-02692-f002:**
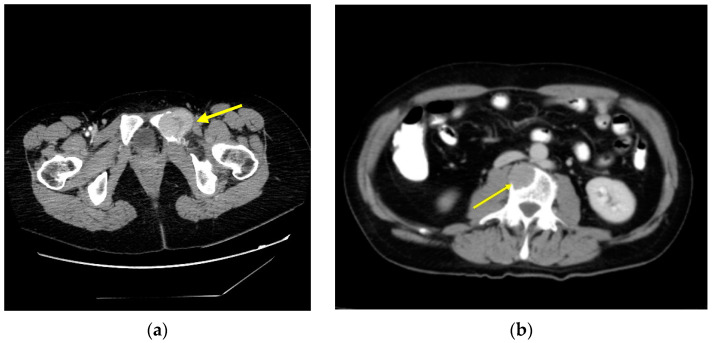
Axial CT scan image: (**a**) bone metastasis of DTC affecting the left ischium (arrow); (**b**) bone metastasis in a lumbar vertebral body (arrow).

**Figure 3 cancers-15-02692-f003:**
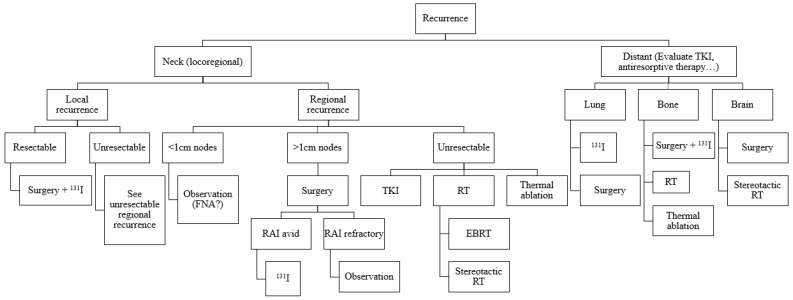
Algorithm for treatment of DTC recurrences.

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
