# Peer review of "Recurrent Differentiated Thyroid Cancer: The Current Treatment Options"

_cancers, 2023, doi:10.3390/cancers15102692_

Round 1

Reviewer 1 Report

Coca-Pelaz and collaborators from the International Head and Neck Scientific Group presented a comprehensive review on the treatment of recurrent differentiated thyroid cancer. The review has good scientific merit and an updated reference list. I have only a few minor suggestions:

(1) A line diagram for diagnostic and treatment algorithms is helpful to the reader.
(2) Most studies report a 65-75% rate of an excellent response after re-operation for neck recurrence (line 205). The authors may expand on some discussions on the role of radioactive iodine therapy in this setting (after redo neck surgery).
(3) Reference 64: may update to NCCN 2022 (10.6004/jnccn.2022.0040).

Reviewer 2 Report

1. The review article provides a brief introduction to the problem of tumor recurrence, which is an important issue in the management of DTC. It also highlights the potential sites of recurrence and the available treatment options, including surgery, radioiodine therapy, external beam radiation therapy, and systemic treatments.

One suggestion I would make is to include some information on the incidence or prevalence of recurrent DTC in the abstract. This would help to provide context for the importance of the topic and the need for effective treatment options. Additionally, it may be helpful to provide a brief overview of the types of systemic treatments that have led to improved progression-free survival, such as targeted therapies and immunotherapies, to give the reader a better understanding of the current landscape of treatment options for recurrent DTC.

2. I would like to request that you consider adding the following reference to the references section of your review article:

Title: "A patent review on efficient strategies for the total synthesis of pazopanib, regorafenib and lenvatinib as novel anti-angiogenesis receptor tyrosine kinase inhibitors for cancer therapy"

This article provides valuable information about the synthesis of pazopanib, lenvatinib, and other anti-angiogenic receptor tyrosine kinase inhibitors that were mentioned in your review article. I believe that this reference will be useful for your readers to understand the chemical structures of these compounds and their potential therapeutic applications.

To enhance the clarity and understanding of the concepts discussed in the review article, it might be helpful to consider adding more figures and/or tables to illustrate the key points. This would make it easier for readers to quickly grasp the information presented and better understand the complex concepts discussed in the article. Additionally, the figures could be designed to highlight important data, such as survival rates or the mechanisms of action of different treatments, to further enhance the impact of the article.
